# Intervertebral Canals and Intracanal Ligaments as New Terms in *Terminologia anatomica*

**DOI:** 10.3390/diagnostics13172809

**Published:** 2023-08-30

**Authors:** Kirill Zhandarov, Ekaterina Blinova, Egor Ogarev, Dmitry Sheptulin, Elizaveta Terekhina, Vladimir Telpukhov, Yuriy Vasil’ev, Mikhail Nelipa, Olesya Kytko, Valery Chilikov, Peter Panyushkin, Olga Drakina, Renata Meilanova, Artem Mirontsev, Denis Shimanovsky, Tatyana Bogoyavlenskaya, Sergey Dydykin, Vladimir Nikolenko, Artem Kashtanov, Vladimir Aliev, Natalia Kireeva, Yulianna Enina

**Affiliations:** 1Department of Operative Surgery and Topographic Anatomy, I.M. Sechenov First Moscow State Medical University, Moscow 119435, Russia; zhandarov_k_a@staff.sechenov.ru (K.Z.);; 2National Medical Research Center of Traumatology and Orthopedics N.N. Pirogova, Moscow 117198, Russia; 3Department of Medical Elementology, Peoples’ Friendship University of Russia, Moscow 117198, Russia; 4Department of Anesthesiology and Intensive Care, I.M. Sechenov First Moscow State Medical University, Moscow 119435, Russia

**Keywords:** intervertebral canal, intracanal ligament, spinal dystrophy, cervical part of the vertebral column dystrophic diseases, cervical part of the vertebral column

## Abstract

This study addresses the cervical part of the vertebral column. Clinical pictures of dystrophic diseases of the cervical part of the vertebral column do not always correspond only to the morphological changes—they may be represented by connective tissue formation and nerve and vessel compression. To find out the possible reason, this morphometric study of the cervical part of the vertebral column in 40 cadavers was performed. CT scans were performed on 17 cadaveric material specimens. A total of 12 histological samples of connective tissue structures located in intervertebral canals (IC) were studied. One such formation, an intracanal ligament (IL) located in the IC, was found. Today, there is no term “intervertebral canal”, nor is there a detailed description of the intervertebral canal in the cervical part of the vertebral column. Cervical intervertebral canals make up five pairs in segments C2–C7. On cadavers, the IC lateral and medial apertures were 0.9–1.5 cm and 0.5–0.9 cm, correspondingly. According to our histological study, the connective tissue structures in the IC are ligaments—IL. According to the presence of these ligaments, ICs were classified into three types. Complete regional anatomy characterization of the IC of the cervical part of the vertebral column with a description of its constituent anatomical elements was provided. The findings demonstrate the need to include the terms “intervertebral canal” and “intervertebral ligament” in the *Terminologia anatomica*.

## 1. Introduction

The clinical manifestation of dystrophic diseases of the cervical part of the vertebral column depends on its anatomical features and pathological changes [1]. In particular, the pathogenesis of radicular syndrome is associated with dystrophy of the intervertebral discs and connective tissue elements of the spine, and of compression of the spinal nerves by fibrous and bony overgrowths [2,3]. At the same time, the clinical manifestations of the disease do not always correspond to the diagnosed morphological changes. The cause of such situations may be due to dense connective tissue formations in the lateral structures of the spine acting as an additional factor for nerve compression [4,5]. Such formations are intracanal ligaments, which were found in previous studies [6,7,8]. But the precise location of these ligaments is still unclear. We assume the presence of a canal in which these ligaments occur.

There is no anatomical term “intervertebral canal” of the cervical part of the vertebral column in the *Terminologia anatomica* [9]. The authors mostly study only the intervertebral foramen and its pathologies [10,11]. The size of the intervertebral foramen plays a big role in dystrophic processes of the spine (central and lateral stenosis, etc.), which leads to the development of not only radicular syndrome but also other pathologies of the nervous and osteoarticular systems [12]. The consequence of these diseases is the impairment of the shoulder girdle and upper extremities functions [13,14]. All this leads to deterioration of the quality of life, disability and, in exceptional cases, the death of the patient [15,16]. Taking into account the diseases mentioned above and general data on dystrophic diseases of the cervical part of the vertebral column, we conducted a study to update knowledge on the morphology of lateral structures of the cervical part of the vertebral column.

On this basis, the aim of our work is to give regional and anatomical characteristics of the intervertebral canal of the cervical part of the vertebral column and to study its soft frame and involvement of the described structures in dystrophic processes by conducting an anatomical, histological and morphometric study of these structures.

## 2. Materials and Methods

### 2.1. Regional Anatomy Study

The materials of this study consist of 40 anatomical specimens of the cervical part of the vertebral column (C-spine), which were extracted from the embalmed bodies of 18 males and 22 females, aged 58–80 years (average age was 69 years). Regional anatomy examination was performed on extracted cadavers’ organs, which included a single block of the cranial base and the cervical part of the vertebral column up to the T1 level.

### 2.2. Methodology of Sectional Study

The embalmment was performed with the use of “Aldofix”. Regional anatomical examination was carried out on cadavers with the soft organs removed by the Shore technique [17]. After dissection of the cervical part of the vertebral column from the surrounding soft tissues, the articulation between the C7 and T1 was separated. The dissected caudal end of the vertebral column was removed anteriorly, and then the entire specimen was extracted along with the preserved base of the skull. Next, deeper dissection was performed within the prevertebral layer of the cervical fascia, and, with the help of a saw, the lateral structures of the cervical part of the vertebral column were cut at an angle of 45° to the sagittal plane. Then, access to the intervertebral canals was opened.

### 2.3. Modeling Technique

To visualize the regional anatomy of the intervertebral canal and to study the relationship between the canal contents and its constituent walls, this anatomical structure was modeled in the vertebral-motor segment of a biological mannequin using 3D-printed models. The walls of the canal were recreated using polymer mass. The vertebral arteries and nerves were modeled with material. Intracanal ligaments and their attachment sites were simulated in the Adobe Photoshop graphical editor by superimposing their images on photographs of cervical vertebrae.

### 2.4. Histological Study

After 24-h fixation of 12 specimens of detected connective tissue structures in 10% formalin solution, the samples were processed using a Leica TP1020 histoprocessor (Germany) according to the standard procedures. Fixed and paraffin-impregnated specimens were embedded in paraffin. Each specimen was then divided into blocks to make 5-micrometer-thick sections. The dewaxed sections were stained with hematoxylin and eosin. The samples were also stained with a mixture of acidic fuchsin and picric acid using the van Gieson method according to the standard protocol. Microscopy of the obtained histological samples was performed using a Leica DM2000 light microscope (Heidelberg, Germany).

### 2.5. Statistical Study

Statistical processing of the data was performed using Microsoft Excel (version 2016, Microsoft, Redmond, Washington, DC, USA). Descriptive statistical methods, including calculation of the arithmetic mean value and standard deviation for the sample, were used.

### 2.6. Morphometric Study and Comparison of Radiological and Anatomical Measurements

The dimensions of the intervertebral canals’ apertures were measured on 40 extracted C-spines containing 200 intervertebral canals (due to the preparation technique, there were only 200 intervertebral canals [17]). All measurements were obtained using a caliper equipped with a vernier scale and a depth gauge with a division value of 0.1 mm. The results were compared with morphometric data on LightSpeed VCT (17 C-spines were studied), where multiplanar reconstruction (MPR) of images in three perpendicular planes with 3D reconstruction of the cervical part of the vertebral column was also performed. Taking into account the paired anatomical material, it was decided to make samples for 17 C-spines for CT and 12 for histological study. Chosen objects were without lifetime or postmortem trauma of the maxillofacial and neck region as well as without pronounced cachexic changes and spine trauma. MPR reconstruction was performed by the software INOBITEC DICOM-viewer version 2.2.0.6264, constructed with Qt 5.5.9 × 64.

## 3. Results

Five pairs of anatomically similar intervertebral canals located in segments C2–C3; C3–C4; C4–C5; C5–C6; and C7–C8 of the cervical part of the vertebral column were identified. Because their regional anatomy showed the morphological structure of the canals, localization of walls and apertures was made. The intervertebral canal resembles a cone with a truncated apex (Figure 1) tapering in the medial direction, with lateral and medial apertures and walls (Figure 2).

The lateral aperture of the intervertebral canal is formed by four walls that include bony, ligamentous and muscular structures. The lower wall is formed by the caudal aspect of the transverse process of the underlying vertebra. The upper is formed by the lateral edge of the transverse process of the superior vertebra; the anterior is formed by the lateral edge of the anterior intertransversarii muscles, which, together with the anterior and middle scalene muscles, is attached to the anterior tubercles of the transverse process of the superior vertebra. The posterior wall is formed by the lateral edge of the posterior intertransversarii muscle, which attaches, together with the posterior scalene muscle and the levator scapulae muscle, to the posterior tubercles of the transverse process of the superior vertebra. The lateral aperture of the intervertebral canal is covered by superficially located structures: fibers of the anterior, middle and posterior scalene muscles and the levator scapulae muscle. The spinal nerve passes through a slit-shaped foramen in the thick part of these muscles. This muscular layer forms the lateral wall of the intervertebral canal.

The medial aperture of the intervertebral canal is formed by the inferior vertebral notch of the superior vertebra and the superior vertebral notch of the inferior vertebra. It also consists of four walls: upper, formed by the upper surface of the transverse process of the superior vertebra; lower, formed by the upper surface of the transverse process of the inferior vertebra; anterior, formed by the anterior intertransversarii muscles and fibers of the longus colli muscle; and posterior, in the formation of which the posterior intertransversarii muscles participate.

The diameter of the lateral and medial apertures and the length of the intervertebral canals were measured. The obtained data are presented in the form of a bar chart (Figure 3).

An oblique sawing through the transverse processes of the cervical vertebrae with an aperture of the intervertebral canal revealed trabeculae with the appearance of a structured soft frame. Histological examination of micropreparations of these structures revealed dense regular fibrous connective tissue consisting of collagen fibers combined into bundles: bundles of the first order were separated by a thin layer of fibroblasts, and bundles of the second order were surrounded by a layer of loose, unformed fibrous connective tissue with blood vessels (Figure 4). Taking into account their organizational structure and anatomical location, the found connective tissue elements were called “intracanal ligaments”. Calcification and ossification processes were expressed in the ligaments (Figure 5); calcified tissues of ligaments were fused with fibrous periosteum and wedged into the structure of spinal nerves (Figure 6), which led to their destruction and fragmentation and to the atrophy of individual nerve fibers (Figure 7).

The intervertebral ligament originates at the base of the transverse process of the upper vertebra, with the upper lateral surface of the lower vertebra (uncovertebral joint) being the other attachment of the ligament. That is, the ligaments cross the intervertebral canal from the upper vertebra to the lower vertebra in the direction from top to bottom and from back to front (Figure 8). In all cases that we evaluated, the ligaments were located above the spinal nerve. However, it is important to emphasize that the ligaments can be located in different directions and planes in relation to the spinal nerve and reduce the free space of the intervertebral canal.

The total number of detected ligaments was counted and the size of the ligament apparatus was measured. According to our study, ligaments were detected in 89% (178/200) of all intervertebral canals; their length ranged from 0.4 cm in C2–C3 segments to 0.6 cm in C6–C7 segments; their width was from 0.15 cm in C2–C3 to 0.3 cm in C6–C7. Distance between the spinal nerve and the intracanal ligament and spinal nerve was 0.9 ± 0.1 cm. All intervertebral canals were subdivided into three types:
Type 1, with clearly expressed, true ligaments (46% of all intervertebral canals);Type 2, where along with true ligaments, there were multiple fibrous growths (43% of all intervertebral canals);Type 3, where ligaments were missing due to fibrous–bony overgrowths (11% of the total number of all intervertebral canals).

We measured the diameters of the intervertebral canal apertures, which on anatomical preparations were 0.9–1.5 cm for the lateral and 0.5–0.9 cm for the medial aperture. The diameters according to MSCT (multispiral computer tomography) data were 0.8–1.4 cm for the lateral and 0.4–0.8 cm for the medial aperture (Figure 9 and Figure 10). When comparing the obtained results, the sizes of the intervertebral canals determined on autopsy material were, in general, larger—up to 1 mm for all dimensions and on all levels. There was no statistically significant relationship between IC size and gender and/or each specific vertebra.

## 4. Discussion

It is well-known that the neck, despite its small size relative to other parts of the body, is an extremely important anatomical region.

The neck also has unique mobility in terms of other anatomical regions. Thanks to the cervical part of the vertebral column, it can bend and unbend. Numerous biomechanical studies show that normal neck flexion is 85° and extension is 70° [18]. The tilt to the right and left has an amplitude of 40° in each direction. In addition, the neck can twist both to the right and to the left. Turns of the head to the right and to the left are 80° in each direction. Such an amplitude of movement complicates the already difficult syntopy of this anatomical region.

Anatomical studies to supplement classical morphological teachings and improve treatment tactics for diseases of the cervical part of the vertebral column are still underway [19,20,21]. However, most authors who made lateral structures of the spine the subject of their scientific research either completely omitted intervertebral canals or their description did not include soft tissues and transverse processes of the vertebrae. In our work, we investigated the structure and all anatomical elements comprising the intervertebral canals of the cervical part of the vertebral column.

In scientific articles, the terms “intervertebral canal” and “intervertebral foramen” were sometimes synonymous [2]. Thorough examination of the lateral structures of the cervical part of the vertebral column made it possible to assess the peculiarities of the shape and size of the transverse processes and intervertebral foramen, to designate the lateral aperture and to describe the anatomical entities involved in the formation of the canal walls. This is the basis for singling out the “intervertebral canal” as an independent anatomical structure and proposing the inclusion of this term in the anatomical nomenclature.

During this study, we identified five pairs of intervertebral canals located in the spinal vertebra-motor segments C2–C3; C3–C4; C4–C5; C5–C6; and C6–C7; the assumption that the number of canals corresponds to the number of spinal nerves was not confirmed, because the first two pass behind the lateral articulations of the atlas and axis and show no evidence of having canals, while the segment C7-T1 is covered by the rib. The size of the canals depends on the level of the segment on which they are located (the lower the canal is, the longer and wider it is) and on the head position.

According to the literature, the space of the groove for the spinal nerve is divided into several zones; it is noted that the medial zone, which includes a complex of such formations as the uncinate process of the cervical vertebrae, the pedicles of the vertebral arch and the zygapophysial joint, plays a greater role in the occurrence of radiculopathy due to its denser environment of bone structures. During our study of the intervertebral canals, we identified connective tissue structures similar to the intervertebral ligaments previously found in the lumbar region [2,22,23]. Having obtained histological confirmation that these elements are indeed part of the ligamentous apparatus and considering their similarity to the above-mentioned structural formations of the lumbar spine, we also decided to call the anatomical formations we found intervertebral ligaments. These ligaments occupy a transverse position over the spinal nerve in the canal and can be true, false or absent due to fibrous–bony overgrowths.

The intervertebral canal of the cervical part of the vertebral column is a functionally important structure because it serves as a protective sheath for the vessels and nerves located there. It is known that narrowing of the medial and lateral intervertebral aperture or the entire intervertebral canal, especially in cases of osteophyte overgrowth, leads to a reduction in the reserve space and squeezing of the roots, nerves, vessels and other anatomical elements passing through them, which promotes the development of severe pathological conditions. This is evidenced, for example, by the occurrence of cervical spondylotic myelopathy or spinal nerve palsy after an anterior cervical discectomy in patients with a narrow intervertebral foramen [24,25]. In addition, the presence of other pathogenetic factors in such patients (hormonal, metabolic or traumatic factors) may cause edema of connective tissue structures and the further narrowing of the already limited space.

It can be assumed that the intervertebral ligament apparatus is also involved in the occurrence of stenosis of the intervertebral canals of the cervical part of the vertebral column, for example, in the case of their fibrosis, which leads to a gradual reduction in the reserve space and squeezing of the spinal nerve segment and vessels by fibrous fascicles of the hypertrophied ligament. A relationship was found between the presence of intrinsic ligaments in the lateral aperture of lumbar intervertebral canals and the development of lumbar segment pathology due to impingement of the underlying neurovascular formations [26], which was not observed in their absence. This fact indirectly confirms our assumptions.

The role of intracanal ligaments as a factor of compressed spinal nerves has not been emphasized in the existing studies. However, opposite theses are put forward, indicating that the ligamentous structures of the intervertebral foramen may play a role in protecting the nerve roots from external traction [27]. Radiating ligaments in the cervical foramina can transfer pulling forces on the nerve roots to the vertebral transverse process, articular facet capsule, vertebral uncinate facet capsule and pedicle, among other structures; such a dispersion of the load on the nerve roots would play a protective role [1,28]. The following transverse approaches to the cervical vertebrae through the anterior neck are recommended: to C3–C4—1 cm below the hyoid bone; to C4–C5—at the level of the upper edge of the thyroid cartilage; to C5–C6—1 cm above the cricoid cartilage; and to C6–C7—1 cm above the sternoclavicular joint. Neurosurgeons also widely use a more universal surgical approach along the anterior edge of the sternocleidomastoid muscle on the right or left. Thus, functionally and surgically, the anterior and posterior sections of the neck are closely interconnected.

Our examination of histological samples of connective tissue structures of intervertebral canals revealed deep tissue dystrophy in the form of ligament calcification, indicating an important role of these ligaments in the dystrophic and adaptive processes taking place in this structure and a predisposition to the development of lateral intervertebral stenosis in people with intervertebral ligamentous apparatus.

Currently, genetic factors influencing the degenerative processes of the spine are being actively studied. There is the opinion that environmental (mechanical and behavioral) factors associated with occupation (physical activity, stress at work) are the main factors in the pathology of the intervertebral disc [28]. Over the years, the classical concept of intervertebral disc aging and wear has been transformed into a complex model of disease with multiple causes based on molecular and genetic changes. Gene mutations have been found to be associated with herniated discs in both humans and animals. Genetic factors appear to play a major role in the pathology of diseases associated with disc degeneration and may be influenced by environmental factors [29]. Obviously, heredity determines, among other things, the stereometry of all anatomical structures. Continuing our research in this direction, the understanding of modern science about the etiology of dystrophic processes around intervertebral canals could be upgraded.

MSCT with MPR and 3D reconstructions can be used to detect intervertebral stenosis and other dystrophic changes in the cervical part of the vertebral column. This method allows us to visualize these structures in maximum detail. A previously published article that reviewed the possibility of using this method for diagnosing these pathological conditions noted the qualitative advantages of MSCT not only over X-ray but even over magnetic resonance imaging (MRI), which provides a low degree of visualization of bone images. We compared radiological and anatomical measurements of the intervertebral canal apertures, which were obtained from examination of anatomical preparations and MSCT. The difference in the obtained results was no more than 1 mm. This can contribute to the diagnosis of cervical stenosis and determination of biological age in forensic examination. Recently, a similar work on the morphometry of the intervertebral canal area using MSCT for preoperative planning in lateral (foraminal) stenosis of the lumbar spine was published [21].

Degenerative processes in the neck lead to pain [30].

Artificial intelligence can be used to draw up a program of therapeutic exercises. At present, there is already experience in using artificial intelligence to develop a twelve-week therapeutic exercise program, which includes one short training session per day [31]. Implementation of the program significantly improved the subjective symptoms of stiff neck, shoulder and lower back pain. In this study, patients showed a high adherence to exercise (92%), which is associated with the use of digital technologies, which helps patients to continue to exercise. In addition, this level of adherence is explained by the improvement in the clinical picture of diseases of the cervical and lumbar spine. Future advances in artificial intelligence are expected to increase systems’ autonomy and reliability, thus providing even more effective tools for the diagnosis and treatment of chronic low back pain [32].

## Figures and Tables

**Figure 1 diagnostics-13-02809-f001:**
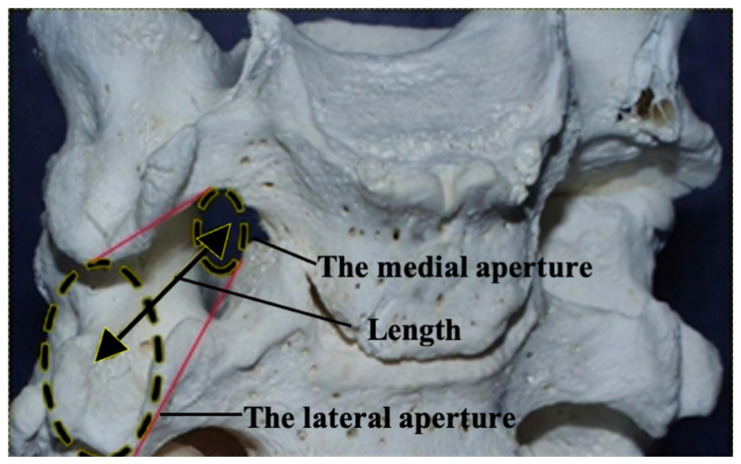
Intervertebral canal of the cervical part of the vertebral column.

**Figure 2 diagnostics-13-02809-f002:**
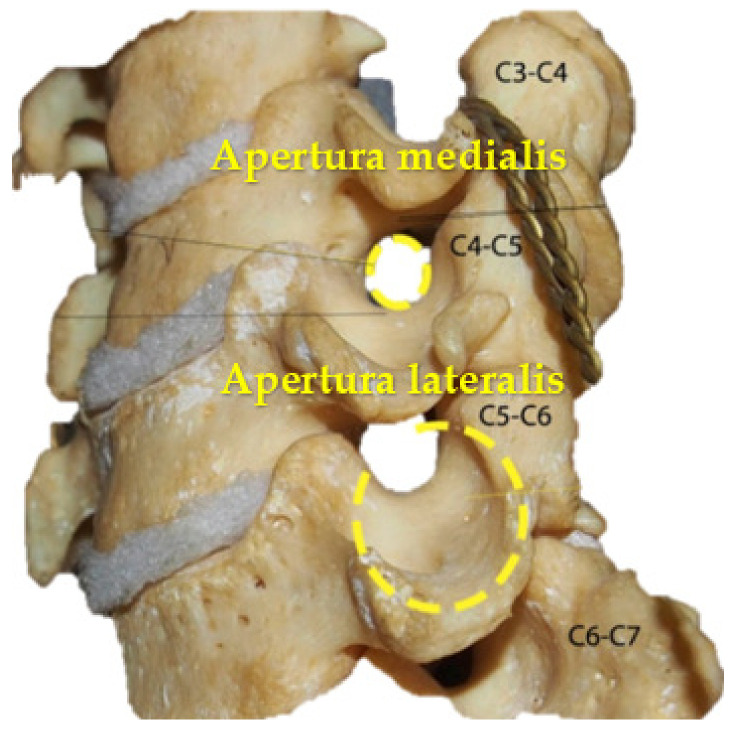
Medial and lateral apertures of the intervertebral canal of the cervical part of the vertebral column.

**Figure 3 diagnostics-13-02809-f003:**
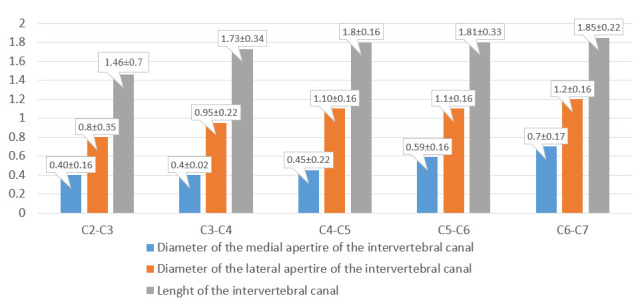
Diameters of medial and lateral apertures and length of intervertebral canals of the cervical part of the vertebral column (average value).

**Figure 4 diagnostics-13-02809-f004:**
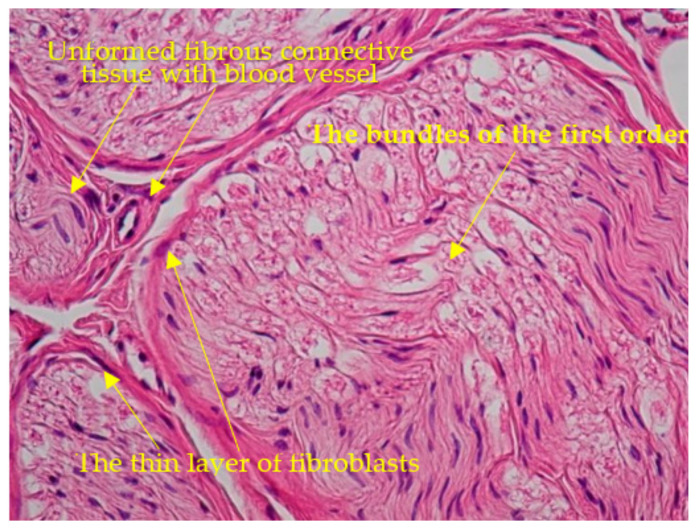
Bundles of the first order. Hematoxylin-eosin staining ×100.

**Figure 5 diagnostics-13-02809-f005:**
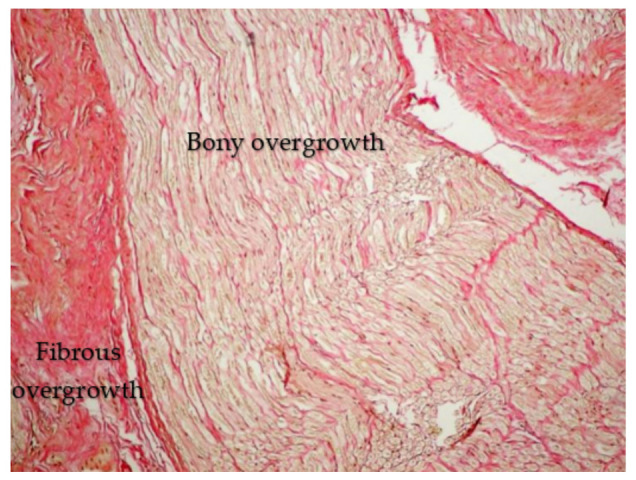
Fibrous–bony overgrowth with compression of the spinal nerve C5. Van Gieson staining ×100.

**Figure 6 diagnostics-13-02809-f006:**
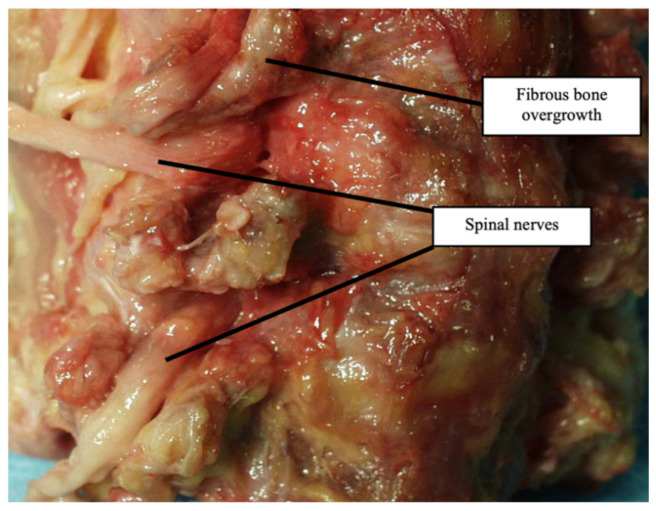
Vertebral osteophytes (stenosis of the intervertebral canal C5–C6, C6–C7).

**Figure 7 diagnostics-13-02809-f007:**
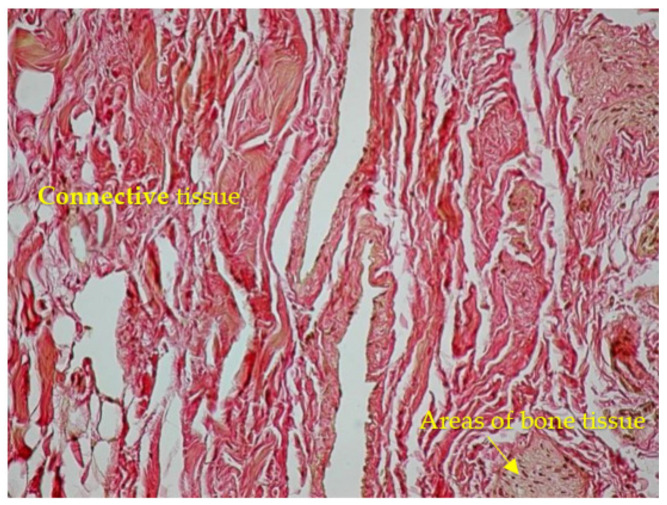
Connective tissue interlayers of the epineurium with areas of bone tissue. Van Gieson stain ×100.

**Figure 8 diagnostics-13-02809-f008:**
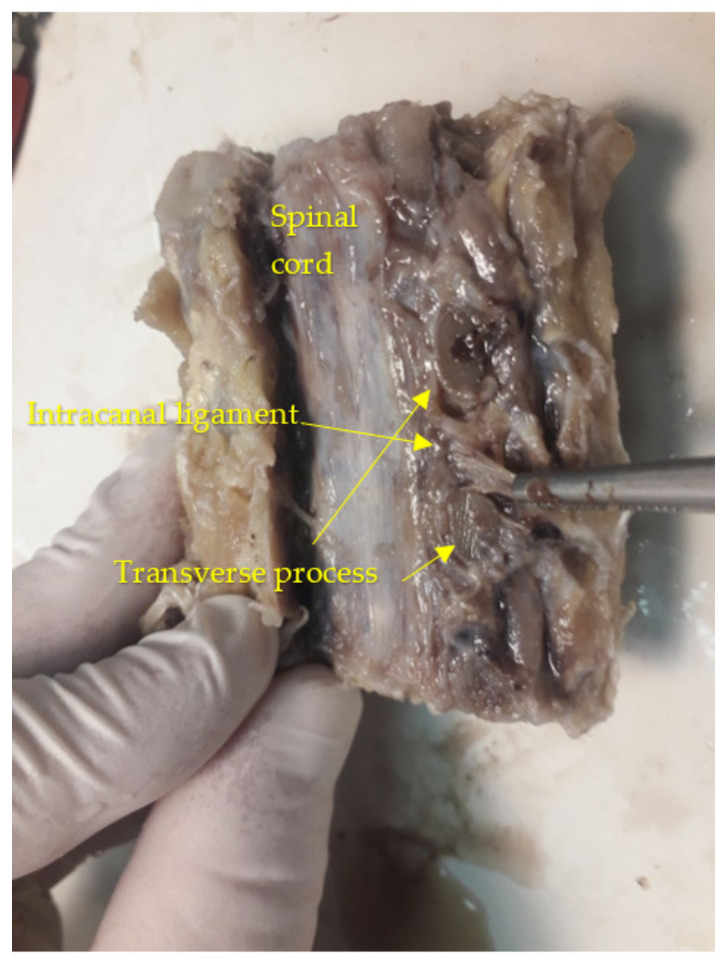
Anatomical preparation of the cervical part of the vertebral column (level C2–C3).

**Figure 9 diagnostics-13-02809-f009:**
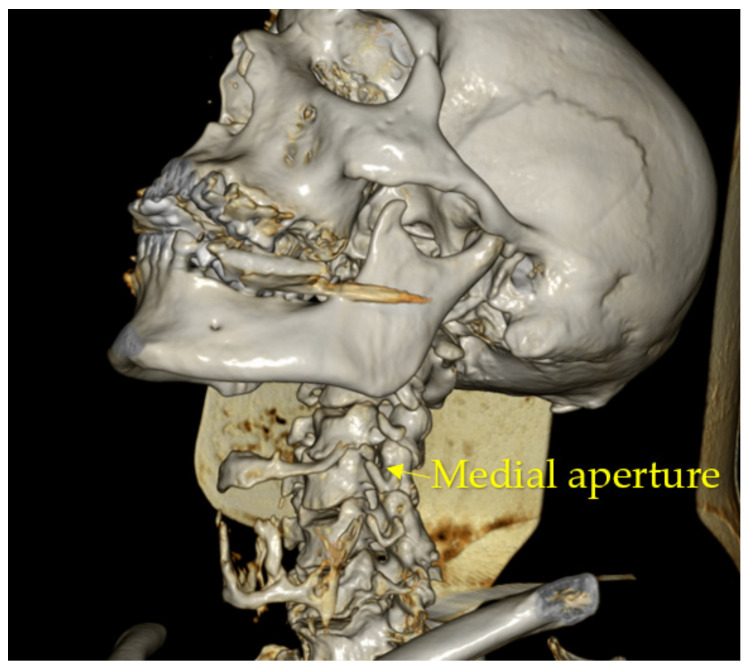
MPR of studied material. Medial aperture.

**Figure 10 diagnostics-13-02809-f010:**
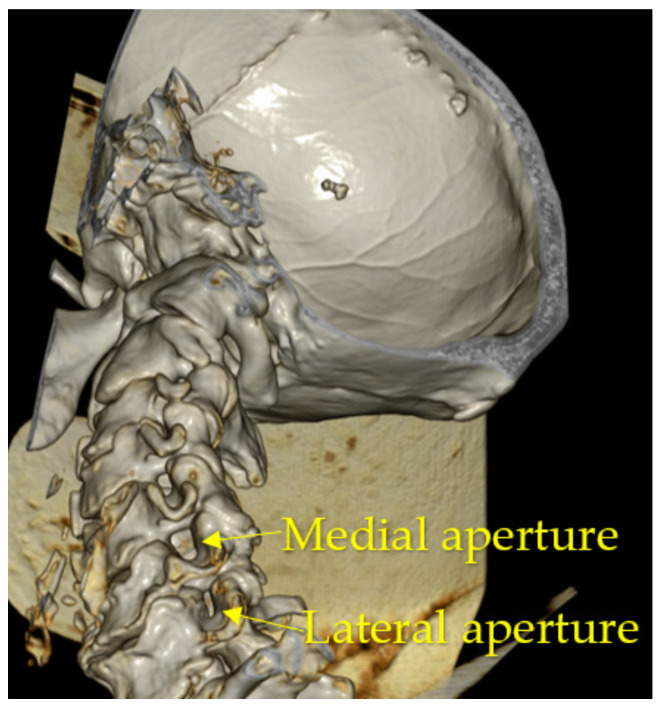
MPR of studied material. Medial and lateral apertures.

## Data Availability

Data are available upon request.

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
