# Peer review of "Intervertebral Canals and Intracanal Ligaments as New Terms in Terminologia anatomica"

_diagnostics, 2023, doi:10.3390/diagnostics13172809_

Round 1

Reviewer 1 Report

The language of the manuscript is poor and has to be checked and ameliorated by a native speaker.
Title and aim: The manuscript handles two topics – one of them is the establishment of the official anatomical term „intervertebral canal“ – as stated at line 226 „This is the basis
for singling out the "intervertebral canal" as an independent anatomical structure and pro- 224
posing the inclusion of this term in the anatomical nomenclature.“ – the other being its morphometric parameters and presence of the intraforaminal ligaments. That is why it is necessary to define the aims and redress the title.
Abstract contains no obvious aim.
Materials – no data on source of cadavers. This must be completed!
Methods – explain how you get only 200 intervetebral foramina in 40 specimens of the cervical part of the vertebral column? Which level was involved/missing. Identical in all specimens?
Methods – missing embalming procedure.
Methods – the modelling technique is unclear and cannot be repeated by other authors according to this poor receipt.
Results – I do not agree that the upper and lower wall of the lateral opening of the „canal“ is formed by the lateral edge of the transverse process – when looking at Figure 2, I suppose, it is the cranial and caudal aspect of the transverse process, respectively. Further, there is no picture providing any proof of the anterior and posterior walls of both the lateral and medial openings – add it.
Results – The dimensions of the „canals“ are not correlated to the size (height) of each specific vertebra and no gender and side differences are presented.
Results – what was the distance between the spinal nerve and the intraforaminal ligament?
Results – what was the incidence at specific levels? What was the distance from the walls? What was the incidence of the three specific types at specific levels? And what was the difference concerning sides and genders?
Results – what were the differences between cadaveric and CT results in each specific case? Identical differences? And for all levels?
Results – no CT shown. Add please. And what about 3D reconstruction of the canal? That should be the most important anatomical conclusion to show the new routine way of clinical examination in case of stenosis/dystrophy.

Generally, change „cervical spine“ to „cervical part of the vertebral column“ throughout the whole text to follow the Terminologia anatomica. Also, unify usage of the term upper/superior and lower/inferior for the vertebrae.
19 – „depends of“ change to „depends on“.
20 – „Clinical pictures don’t“ change to „Clinical pictures do not“.
22 – „of a such formations“ change to „of such formations“.
22 – „intervertebral canals“ – do you mean intervetebral foramina? Or something else? How does your „canal“ corresponds to „foramen“? Explain and change accordingly throughout the whole text.
23 – „Today there are no term "intervertebral canal" (IC)“ – redress.
I stopped checking the grammar and style and highly recommend language correction by native speaker.
24 – „anatomical biological models of cervical spine“ – what is it?
27 – „IC foramen were 0.9-1.5 cm for the lateral and 0.5-0.9 cm for the medial one.“ – this does not make sense.
34 – „International Anatomical Nomenclature“ – do you mean Terminologia Anatomica?
38 – „outcome of dystrophic diseases“ – the outcome of a disease?
46 – „hernia“ change to „heniation“.
48 – „Such formations are intraforaminal ligaments found in previous studies.“ – I miss here relevnt references. Add.
50 – „in intervertebral openings – in the so-called intervertebral canals“ – these structures are called „intervertebral foramina“.
53 – „the walls of the intervertebral foramen“ – they should be defined here.
58 – „of various organs“ – specify, of which organs?
71 – „preparations“ change to „specimen“.
75 – „Th1 level“ change to „T1 level“.
80 – „separation of the articulation between C7 and Th1 vertebrae was performed“ – what does this mean?
81 – „distal“ change to „caudal“.
82 – „Layer-by-layer dissection within the prevertebral fascia“ – First, correct term is „prevertebral layer of cervical fascia“. Second a „layer-by-layer dissection“ cannot be peformed within one layer = fascia. Do you mean „deeper/posterior to“?
88 – „vertebral-motor segment of a biological mannequin“ – explain what do you mean by this and how have you modelled it?
90 – „simulated material“ – what is it?
107 – „c-„ change to „C-„.
110  - „17 c-spines were studied“ – how have you selected the 17 out if the 40? And why 17?
116 – „topographic“ change to „topographical“.
Unify terms „orifice, aperture, foramen, opening“.
129 – which „intertransversarii muscles“? Specify.
Figure 1 and 2 – unify the labels language.
Figure 3 – refine the incorrect labels and unify their language; redress and specify „intertransversarii, deep cervical muscles (forming the walls of intervertebral canals)“.
Figure 4 – what are the values shown? Mean, average, media nor what?
161 – „endothenium“ - ?
164 – „marginal ligaments“ – what do you man by that?
Figure 5 and 7 are not labelled.
170 – „Van Gizon“ change to „Van Gieson“.
178 – „Luschka's joints“ – explain or use a proper term („uncovertebral joint“).
179 – „the second point of attachment“ change to „the opposite/other attachment“.
Figure 8 – label more structures, and also the spinal nerve, further 1 – is not „intervertebral joint“ but „zygapophysial joint“.
190 – „in 89% of all vertebral canals“ – do you the intervertebral canals? And of all - do you mean all 12 (histological study) or 27 (CT study)? Add the exact numbers in brackets, e.g. 89% (15/17).
194 – „fused false ligaments“ – what does it mean? Why not shown on a histological slide?
204-5 – delete this redundant sentence.
206-11 – also, delete this paragraph being out of topic.
226 – what do you mean by „motor segments C2-C3; C3-C4; C4-C5; C5-C6 and C6-C7“? change these to usual terms and definitions.
227 – „the assumption that the number of canals corresponds to the number of spinal nerves was not confirmed, because the first two pass behind the lateral articulations of the atlant and epistrophy“ – the authors have not proved this that is why the cannot conclude it!
227 – „atlant and epistrophy“ change to „atlas and axis“.
230 – „the segment C7-Th1 is covered by the rib“ – and it means that there is no canal? This does not make sense unless proved!
232 – „and on the head position“ – either add reference or own author’s data.
233 – „neural sulcus space“ change to „space of the groove for the spinal nerve“.
235 – „facet joints“ change to „zygapophysial joint“.
265 – „emphasized in the studies of several authors. However, in several works, opposite theses“ – no references for the former and only one for the latter is not „several“ – please, complete or delete.
271 – „Ligaments exert the strongest protective effect on the C5 nerve roots“ – missing reference.
277-286 – delete this redundant and off-topic paragraph.
293 – „A number of authors argue“ and one reference – complete or delete.
296 – „IVD“ – unexplained abbreviation.
306 – „A previously published article“ – missing reference.
326 – „The results of this study also indicate that poor posture or incorrect patterns of movement of the cervical spine can reduce the area of the intervertebral foramen“ – how the study indicates that? The study brings passive data only. Delete this too self-confident sentence.
329-331 – missing reference.
333-343 – delete this off-topic paragraph.
References – not all references are well formatted. Check them carefully and unify the format!

See above.

Author Response

Dear Reviewer 1,
Thank you for your comments. Further we provide point-by point response on them:

The language of the manuscript is poor and has to be checked and ameliorated by a native speaker.

We have re-read and particularly re-write our article text to make clearer.

Title and aim: The manuscript handles two topics – one of them is the establishment of the official anatomical term „intervertebral canal“ – as stated at line 226 „This is the basis
for singling out the "intervertebral canal" as an independent anatomical structure and pro- 224
posing the inclusion of this term in the anatomical nomenclature.“ – the other being its morphometric parameters and presence of the intraforaminal ligaments. That is why it is necessary to define the aims and redress the title.

Our new title is: “Intervertebral canals and intracanal ligaments as new terms in Termologia Anatomica.” Thus, our title corresponds to our study’ aim.

Abstract contains no obvious aim.

We have point-out our aim more clearly in the abstract.

Materials – no data on source of cadavers. This must be completed!

Cadaver material was taken from the Department cadavers base.
Methods – explain how you get only 200 intervetebral foramina in 40 specimens of the cervical part of the vertebral column? Which level was involved/missing. Identical in all specimens?

Such number of the intervertebral canals is caused by preparation technique. We have mentioned it in the text.

Methods – missing embalming procedure.

The embalmed was performed with usage of ““Aldofix” (https://novochem.ru/upload/iblock/26f/mw3atwj9pdrr57epffegs8byskleieg3/Aldofix_specification_eng.pdf)

Results – I do not agree that the upper and lower wall of the lateral opening of the „canal“ is formed by the lateral edge of the transverse process – when looking at Figure 2, I suppose, it is the cranial and caudal aspect of the transverse process, respectively. Further, there is no picture providing any proof of the anterior and posterior walls of both the lateral and medial openings – add it.

We added Fig. 10 and 11 and corrected our description.

Results – The dimensions of the „canals“ are not correlated to the size (height) of each specific vertebra and no gender and side differences are presented.

No statistically significant relationships were found. Thus, we have not mention this into the main text.

Results – what was the distance between the spinal nerve and the intraforaminal ligament?

It was near 0.8-1 cm according to our measurement.

Results – what was the incidence at specific levels? What was the distance from the walls? What was the incidence of the three specific types at specific levels? And what was the difference concerning sides and genders?

Results – what were the differences between cadaveric and CT results in each specific case? Identical differences? And for all levels?

“When comparing the obtained results, the sizes of the intervertebral canals determined on autopsy material were, in general, larger - up to 1 mm at all dimensions and on all levels.” – we added this sentence to the text.

Results – no CT shown. Add please. And what about 3D reconstruction of the canal? That should be the most important anatomical conclusion to show the new routine way of clinical examination in case of stenosis/dystrophy.

We have added MPR – Fig.10 and 11.

Generally, change „cervical spine“ to „cervical part of the vertebral column“ throughout the whole text to follow the Terminologia anatomica.

We have checked it and corrected.

Also, unify usage of the term upper/superior and lower/inferior for the vertebrae.
19 – „depends of“ change to „depends on“.

We have deleted this sentence.

20 – „Clinical pictures don’t“ change to „Clinical pictures do not“.

We have checked it and corrected.

22 – „of a such formations“ change to „of such formations“.

We have checked it and corrected.

22 – „intervertebral canals“ – do you mean intervetebral foramina? Or something else? How does your „canal“ corresponds to „foramen“? Explain and change accordingly throughout the whole text.

Thankyou for your correction. There is term  “intervetebral foramina” and there is no term “intervertebral canal”. We find intervertebral canals in our study and point it to be included in Terminologia Anatomica.

23 – „Today there are no term "intervertebral canal" (IC)“ – redress.

We have checked it and corrected.

I stopped checking the grammar and style and highly recommend language correction by native speaker.
24 – „anatomical biological models of cervical spine“ – what is it?

It is cadaver material. We have change it.

27 – „IC foramen were 0.9-1.5 cm for the lateral and 0.5-0.9 cm for the medial one.“ – this does not make sense.

We have checked it and corrected.

34 – „International Anatomical Nomenclature“ – do you mean Terminologia Anatomica?

We have checked it and corrected.

38 – „outcome of dystrophic diseases“ – the outcome of a disease?

We have deleted it.

46 – „hernia“ change to „heniation“.

We have deleted it.

48 – „Such formations are intraforaminal ligaments found in previous studies.“ – I miss here relevnt references. Add.

We have checked added 3 studies on this topic.

50 – „in intervertebral openings – in the so-called intervertebral canals“ – these structures are called „intervertebral foramina“.

We have checked it and corrected.

53 – „the walls of the intervertebral foramen“ – they should be defined here.

We have completely re-write this sentence to make it more clear and add reference to prove our words.

58 – „of various organs“ – specify, of which organs?

We have deleted it.

71 – „preparations“ change to „specimen“.

We have checked it and corrected.
75 – „T1 level“ change to „T1 level“.

We have checked it and corrected.
80 – „separation of the articulation between C7 and Th1 vertebrae was performed“ – what does this mean?
81 – „caudal“ change to „caudal“.
82 – „Layer-by-layer dissection within the prevertebral fascia“ – First, correct term is „prevertebral layer of cervical fascia“. Second a „layer-by-layer dissection“ cannot be peformed within one layer = fascia. Do you mean „deeper/posterior to“?

We have completely re-write the “ Methodology of sectional study” section.

88 – „vertebral-motor segment of a biological mannequin“ – explain what do you mean by this and how have you modelled it?

The vertebral-motor segment is part of the spine, which consists of two adjacent vertebrae.

90 – „simulated material“ – what is it?

Its special material for 3D modeling.

107 – „c-„ change to „C-„.

We have checked it and corrected.

110  - „17 c-spines were studied“ – how have you selected the 17 out if the 40? And why 17?

"By Hanspach L., et al., 2021 it is known that in research involving healthy peoples the statistics should be limited by small sample sizes from 1 to 10. But cadaver material is not accessible for the reasons not related to authors [18]. Taking it into account and paired anatomical material it was decided to make sample for 17 C-spines for CT and 12 for histological study. Chosen objects were without a lifetime and postmortem trauma of the maxillofacial and neck region, as well as without pronounced cachexic changes and spine trauma" we have added this paragraph to the "Morphometric study and comparison of radiological and anatomical measurements" section.

116 – „topographic“ change to „topographical“.

We change it to “regional-anatomy”

Unify terms „orifice, aperture, foramen, opening“.

We have checked it and corrected.

129 – which „intertransversarii muscles“? Specify.

We have checked it and corrected.

Figure 1 and 2 – unify the labels language.

We have checked it and corrected.

Figure 3 – refine the incorrect labels and unify their language; redress and specify „intertransversarii, deep cervical muscles (forming the walls of intervertebral canals)“.

We have removed this figure.
Figure 4 – what are the values shown? Mean, average, media nor what?

We have checked it and corrected.

161 – „endothenium“ - ?

We have deleted it.

164 – „marginal ligaments“ – what do you man by that?

We have deleted it.

Figure 5 and 7 are not labelled.

We have checked it and corrected

170 – „Van Gizon“ change to „Van Gieson“.

We have checked it and corrected.

178 – „Luschka's joints“ – explain or use a proper term („uncovertebral joint“).

We have checked it and corrected.

179 – „the second point of attachment“ change to „the opposite/other attachment“.

We have checked it and corrected.

Figure 8 – label more structures, and also the spinal nerve, further 1 – is not „intervertebral joint“ but „zygapophysial joint“.

We have upload new explanations to the figures.

We have checked it and corrected
190 – „in 89% of all vertebral canals“ – do you the intervertebral canals? And of all - do you mean all 12 (histological study) or 27 (CT study)? Add the exact numbers in brackets, e.g. 89% (15/17).
194 – „fused false ligaments“ – what does it mean? Why not shown on a histological slide?

We have deleted it.

204-5 – delete this redundant sentence.
206-11 – also, delete this paragraph being out of topic.

We have deleted it.

226 – what do you mean by „motor segments C2-C3; C3-C4; C4-C5; C5-C6 and C6-C7“? change these to usual terms and definitions.

We have checked it and corrected.

227 – „the assumption that the number of canals corresponds to the number of spinal nerves was not confirmed, because the first two pass behind the lateral articulations of the atlant and epistrophy“ – the authors have not proved this that is why the cannot conclude it!
227 – „atlant and epistrophy“ change to „atlas and axis“.

We have checked it and corrected.

230 – „the segment C7-Th1 is covered by the rib“ – and it means that there is no canal? This does not make sense unless proved!
232 – „and on the head position“ – either add reference or own author’s data.

233 – „neural sulcus space“ change to „space of the groove for the spinal nerve“.

We have checked it and corrected.

235 – „facet joints“ change to „zygapophysial joint“.

We have checked it and corrected .

265 – „emphasized in the studies of several authors. However, in several works, opposite theses“ – no references for the former and only one for the latter is not „several“ – please, complete or delete.

We have checked it and corrected

271 – „Ligaments exert the strongest protective effect on the C5 nerve roots“ – missing reference.

We have checked it and corrected

277-286 – delete this redundant and off-topic paragraph.
We have checked it and corrected

293 – „A number of authors argue“ and one reference – complete or delete.

We have checked it and corrected

296 – „IVD“ – unexplained abbreviation.

We have checked it and corrected. IVD is intervertebral disk.

306 – „A previously published article“ – missing reference.
326 – „The results of this study also indicate that poor posture or incorrect patterns of movement of the cervical spine can reduce the area of the intervertebral foramen“ – how the study indicates that? The study brings passive data only. Delete this too self-confident sentence.
329-331 – missing reference.
333-343 – delete this off-topic paragraph.

We have checked it and corrected

References – not all references are well formatted. Check them carefully and unify the format!
We have upload new list of references and made it in accordance with the journal recommendation.

Reviewer 2 Report

Re: Manuscript ID: diagnostics-2508350

This is an article dealing with an anatomical issue. I reported different comments to improve the paper.

Line 49. Which are the “previous studies”?

Lines 114-115. The authors stated that only “5 pairs of anatomically similar intervertebral canals located in segments C2-C3; C3-C4; C4-C5; C5-C6 and C7-C8 of the cervical spine were identified”. Considering that 40 preparations were used, 400 intervertebral canals were available. The authors studied only 200 intervertebral canals. This means that only one side was examined. Which one? Why this choice? A comparison of both sides would provide more information.

Fig. 2. “incisura vertebralis superior” and “incisura vertebralis inferior” must be switched. They refer to the vertebra not to the apertura medialis.

Fig. 3. Replace “intertransarii” withintertransversarii” and coli” with colli”.

The authors are invited to standardize the nomenclature of the intervertebral canals: for example, apertura lateralis or lateral opening or lateral foramen?

The intervertebral canal, which has a length, is formed by walls, whereas the foramen, which has not a length, is formed by margins. Please, correct.

Fig. 4. The quality of this figure is poor and the reviewer was not able to read the numbers.

Lines 162-163. According to the main purpose of the authors, I would replace “intraforaminal ligaments” (which foramen?) with “intracanal ligaments” (as described in lines 179-180).

Figs. 5, 6, 8. Please, add arrows or symbols to better illustrate the picture with a more detailed caption.

Line 304. Explanation of the acronym MSCT must be anticipated in the line 200.

Line 296. IVD (?).

Line 307. Which article?

Author Response

Dear Reviewer,
Thank you for your comments. Further we provide point-by point answer on them:

This is an article dealing with an anatomical issue. I reported different comments to improve the paper.

Line 49. Which are the “previous studies”?

We have added appropriate studies.

Lines 114-115. The authors stated that only “5 pairs of anatomically similar intervertebral canals located in segments C2-C3; C3-C4; C4-C5; C5-C6 and C7-C8 of the cervical spine were identified”. Considering that 40 preparations were used, 400 intervertebral canals were available. The authors studied only 200 intervertebral canals. This means that only one side was examined. Which one? Why this choice? A comparison of both sides would provide more information.

Such number of the intervertebral canals is caused by preparation technique. We have mentioned it in the text.

Fig. 2. “incisura vertebralis superior” and “incisura vertebralis inferior” must be switched. They refer to the vertebra not to the apertura medialis.

We have checked it and corrected.

Fig. 3. Replace “intertransarii” with “intertransversarii” and “coli” with “colli”.
We have deleted this figure.

The authors are invited to standardize the nomenclature of the intervertebral canals: for example, apertura lateralis or lateral opening or lateral foramen?

We have checked it and corrected

The intervertebral canal, which has a length, is formed by walls, whereas the foramen, which has not a length, is formed by margins. Please, correct.

Fig. 4. The quality of this figure is poor and the reviewer was not able to read the numbers.

We have checked it and corrected

Lines 162-163. According to the main purpose of the authors, I would replace “intraforaminal ligaments” (which foramen?) with “intracanal ligaments” (as described in lines 179-180).

Thankyou for your idea. We strongly agree with you.

Figs. 5, 6, 8. Please, add arrows or symbols to better illustrate the picture with a more detailed caption.

We have checked it and corrected

Line 304. Explanation of the acronym MSCT must be anticipated in the line 200.

We have checked it and corrected

Line 296. IVD (?).

We have checked it and corrected. IVD is intervertebral disk.

Line 307. Which article?

We have checked it and corrected

Round 2

Reviewer 1 Report

There are several points which have not been fully or partially addressed:
1.    “The language of the manuscript is poor and has to be checked and ameliorated by a native speaker.” I suppose the manuscript was not checked by a native speaker as I have found many new mistakes. I insist on a review by a native speaker who should be then acknowledged either as co-author or at least in the Acknowledgement section.
2.    “Materials – no data on source of cadavers. This must be completed!” The answer “Cadaver material was taken from the Department cadavers base” is not sufficient. What is the source of the base? Are these bodies donors with signed consent or different?
3.    “Methods – missing embalming procedure.” The answer “The embalmed was performed with usage of ““Aldofix”” is only in the File “Answer to Reviewers Comments” but not in the manuscript. Please, add it there!
4.    “Results – The dimensions of the „canals“ are not correlated to the size (height) of each specific vertebra and no gender and side differences are presented.” The answer “No statistically significant relationships were found. Thus, we have not mention this into the main text.” must be stated in the text otherwise the reader will not know there and could be confused. Add it there, please!
5.    “Results – what was the distance between the spinal nerve and the intraforaminal ligament?” The answer “It was near 0.8-1 cm according to our measurement” is not in the manuscript. Please, add it!
6.    “Results –What was the incidence of the three specific types at specific levels? And what was the difference concerning sides and genders?” – Not added, I still miss this information!

Further, the work by Hanspach et al. (2021) is not relevant as it relates to MRI studies and its results and opinion cannot be translated to pure anatomical study.
Minor new comments:
27 – „ of 40 cadavers cervical part of the vertebral column was performed“ change to „of the cervical part of the vertebral column in 40 cadaver was performed“.
29 – unexplained abbreviation: „IC“ change to „intervertebral canals (IC)“.
61  „But its 61 still unclear place of such ligaments. We assume the presence of a channel in which this 62 ligaments lies“ change to „But the precise location of these ligaments is still unclear. We assume the presence of a canal in which this“.
111 – „Topographic“ change to „Topographical“.
111 – „corpses“ change to „cadavers“.
114 – „prepared distal end of the spine“ change to „dissected caudal end of the vertebral column“.
116 – „of cervical fascia“ change to „of the cervical fascia“.
117 – „cervical spine“ change to „cervical part of the vertebral column“.
122 – „wuth“ change to „with“.
150-153 – delete this unrelated new text.
160 – „seg-ments“ change to „segments“.
161 – „cervical spinecervical part of the ver-tebral column“ change to „cervical part of the vertebral column“.
Figure 5 labels:
„Fibrous“ change to „Fibrous overgrowth“
Figure 8 labels:
Replace „spine“ with more relevant term.

I suppose the manuscript was not checked by a native speaker as I have found many new mistakes. I insist on a review by a native speaker who should be then acknowledged either as co-author or at least in the Acknowledgement section.

Author Response

Dear Reviewer,
Thank you for your big role in our article proceedings and your notes that make your article more clear for readers. There are our corrections on your points below:

There are several points which have not been fully or partially addressed:

  1. “The language of the manuscript is poor and has to be checked and ameliorated by a native speaker.” I suppose the manuscript was not checked by a native speaker as I have found many new mistakes. I insist on a review by a native speaker who should be then acknowledged either as co-author or at least in the Acknowledgement section.

Our manuscript was checked by Daniel Gitelson (we add his certificate) and pointed him in “Acknowledgement” section.

  1. “Materials – no data on source of cadavers. This must be completed!” The answer “Cadaver material was taken from the Department cadavers base” is not sufficient. What is the source of the base? Are these bodies donors with signed consent or different?

Department cadaver base is formed from University cadaver base. University gets cadaver from hospitals, clinic and etc with all necessary consents signed by patient/ relatives or special delegation.

  1. “Methods – missing embalming procedure.” The answer “The embalmed was performed with usage of ““Aldofix”” is only in the File “Answer to Reviewers Comments” but not in the manuscript. Please, add it there!

We have followed to your recommendation and added it.

  1. “Results – The dimensions of the „canals“ are not correlated to the size (height) of each specific vertebra and no gender and side differences are presented.” The answer “No statistically significant relationships were found. Thus, we have not mention this into the main text.” must be stated in the text otherwise the reader will not know there and could be confused. Add it there, please!

We have followed to your recommendation and added it.

  1. “Results – what was the distance between the spinal nerve and the intraforaminal ligament?” The answer “It was near 0.8-1 cm according to our measurement” is not in the manuscript. Please, add it!

We have followed to your recommendation and added it.

  1. “Results –What was the incidence of the three specific types at specific levels? And what was the difference concerning sides and genders?” – Not added, I still miss this information!

We have checked it and added.

Further, the work by Hanspach et al. (2021) is not relevant as it relates to MRI studies and its results and opinion cannot be translated to pure anatomical study.

Minor new comments:

27 – „ of 40 cadavers cervical part of the vertebral column was performed“ change to „of the cervical part of the vertebral column in 40 cadaver was performed“.

We have checked it and corrected in accordance with your recommendation.

29 – unexplained abbreviation: „IC“ change to „intervertebral canals (IC)“.

We have checked it and corrected in accordance with your recommendation.

61  „But its 61 still unclear place of such ligaments. We assume the presence of a channel in which this 62 ligaments lies“ change to „But the precise location of these ligaments is still unclear. We assume the presence of a canal in which this“.

We have checked it and corrected in accordance with your recommendation.

111 – „Topographic“ change to „Topographical“.

We have checked it and change to “regional”.

111 – „corpses“ change to „cadavers“.

We have checked it and corrected in accordance with your recommendation.

114 – „prepared distal end of the spine“ change to „dissected caudal end of the vertebral column“.

We have checked it and corrected in accordance with your recommendation

116 – „of cervical fascia“ change to „of the cervical fascia“.

We have checked it and corrected in accordance with your recommendation

117 – „cervical spine“ change to „cervical part of the vertebral column“.

We have checked it and corrected in accordance with your recommendation

122 – „wuth“ change to „with“.

We have checked it and corrected in accordance with your recommendation

150-153 – delete this unrelated new text.

We have deleted it and corrected in accordance with your recommendation

160 – „seg-ments“ change to „segments“.

We have checked it and corrected in accordance with your recommendation

161 – „cervical spinecervical part of the ver-tebral column“ change to „cervical part of the vertebral column“.

We have checked it and corrected in accordance with your recommendation

Figure 5 labels:

„Fibrous“ change to „Fibrous overgrowth“

We have followed to your recommendation and completed figure

Figure 8 labels:

Replace „spine“ with more relevant term.

 We have followed to your recommendation and completed figure

Reviewer 2 Report

The authors provided a satisfactory revision of their manuscript.

Author Response

Dear Reviewer,
Thank you for your high-quality review of our work.